# Shade-induced nuclear localization of PIF7 is regulated by phosphorylation and 14-3-3 proteins in *Arabidopsis*

**Xu Huang[1,2], Qian Zhang[1,2], Yupei Jiang[1,2], Chuanwei Yang[1,2], Qianyue Wang[1,2], Lin Li[1,2]\***

[1]State Key Laboratory of Genetic Engineering, School of Life Sciences, Fudan University, Shanghai, China; [2]Institute of Plant Biology, School of Life Sciences, Fudan University, Shanghai, China

**Abstract** Shade avoidance syndrome enables shaded plants to grow and compete effectively against their neighbors. In *Arabidopsis*, the shade-induced de-phosphorylation of the transcription factor PIF7 (PHYTOCHROME-INTERACTING FACTOR 7) is the key event linking light perception to stem elongation. However, the mechanism through which phosphorylation regulates the activity of PIF7 is unclear. Here, we show that shade light induces the de-phosphorylation and nuclear accumulation of PIF7. Phosphorylation-resistant site mutations in PIF7 result in increased nuclear localization and shade-induced gene expression, and consequently augment hypocotyl elongation. PIF7 interacts with 14-3-3 proteins. Blocking the interaction between PIF7 and 14-3-3 proteins or reducing the expression of 14-3-3 proteins accelerates shade-induced nuclear localization and de-phosphorylation of PIF7, and enhances the shade phenotype. By contrast, the 14-3-3 overexpressing line displays an attenuated shade phenotype. These studies demonstrate a phosphorylation-dependent translocation of PIF7 when plants are in shade and a novel mechanism involving 14-3-3 proteins, mediated by the retention of PIF7 in the cytoplasm that suppresses the shade response.
DOI: https://doi.org/10.7554/eLife.31636.001

\*For correspondence:
linli@fudan.edu.cn

**Competing interests:** The authors declare that no competing interests exist.

## Introduction

Because chlorophyll preferentially absorbs light in the red and blue ranges but not in the far-red range of the light spectrum, a perceived decrease in the ratio of red/far-red (R/FR) radiation, and thus in photosynthetically active radiation (PAR) of between 400 and 700 nm, provides a signal that shading by other plants is imminent. Shade-intolerant plants, such as *Arabidopsis thaliana*, sense this reduction and initiate the shade avoidance syndrome (SAS). In the SAS, energy resources are reallocated from storage organs to stem-like organs, including hypocotyls and petioles, thereby enabling plants to initiate a rapid growth response (*Cole et al., 2011*; *Casal, 2012*). Prolonged shade exposure leads to reduced branching, early flowering and seed set, and reduced yield (*Ballaré, 1999*; *Franklin and Whitelam, 2005*; *Procko et al., 2014*).

The R/FR-absorbing photoreceptor phytochrome B (phyB) plays the most dominant role during the SAS (*Reed et al., 1993*). In an open environment under sunlight (where the ratio of R/FR is about 1.2–1.5), most of the phyB is in the far-red-absorbing (Pfr) active form and moves to the nucleus, where it interacts with basic helix-loop-helix (bHLH) proteins known as PHYTOCHROME-INTERACTING FACTORS (PIFs) (*Duek and Fankhauser, 2005*; *Leivar and Quail, 2011*). The photoactivation of phyB induces the rapid phosphorylation of PIF1/3/4/5 prior to their degradation (*Shen et al., 2005*; *Al-Sady et al., 2006*; *Shen et al., 2007*), although the short half-lives of these PIFs impedes the tracing of the phosphorylated forms. When plants are in the shade, phyB is mostly in the inactive red-

absorbing (Pr) cytosolic form, which facilitates the accumulation of PIFs and restricts the growth of the hypocotyl (*Lorrain et al., 2008*; *Leivar and Quail, 2011*).

PIF7 is a major regulator of the shade response, as shown by the severe shade-defective phenotype of *pif7* mutants (*Li et al., 2012*; *de Wit et al., 2015*; *Mizuno et al., 2015*). PIF7 is less vulnerable than PIF1/3/4/5 to the rapid turnover of induced by the Pfr form of phyB (*Leivar et al., 2008*). Instead, the activity of PIF7 is controlled by rapid de-phosphorylation in response to shade, which leads to its binding to G-boxes in the promoters of auxin biosynthesis genes, causing an increase in auxin levels and a rapid growth response (*Li et al., 2012*).

The 14-3-3 proteins are highly conserved in all eukaryotes. Research in recent years has revealed several putative 14-3-3 targets in plants (*Jaspert et al., 2011*; *Wang et al., 2011*; *Yoon and Kieber, 2013*; *Zhou et al., 2014*). These studies have revealed that 14-3-3 proteins can interact with the phosphorylated forms of their client proteins in response to certain signals, and that this binding finalizes the signaling event by enabling a change in the subcellular localization, protein stability or intrinsic enzymatic activity of the client, or by promoting an association between the client and other proteins. The cellular 14-3-3 'pool' enables these proteins to react to altered signaling cues in an immediate and precise way through dynamic interactions with their clients.

Here, we demonstrate a shade induction of the nuclear localization of dephosphorylated PIF7 and a role for the 14-3-3 proteins in the cytoplasmic retention of PIF7 in *Arabidopsis*. Our work reveals a novel mechanism that rapidly switches PIF7 function in response to light conditions and the role of 14-3-3 proteins in SAS.

## Results

### Shade induces the rapid nuclear localization of PIF7

To monitor the cellular localization of PIF7, we generated 35S::*GFP-PIF7* transgenic plants and analyzed the GFP signal in white-light-grown seedlings before and after shade treatment. Impressively, GFP-PIF7 rapidly accumulated in the nucleus when plants were placed in the shade, as observed in the cotyledon and the hypocotyl of transgenic lines. The extent of this shade response decreased gradually from the top to the bottom of the hypocotyls (*Figure 1—figure supplement 1a*). At the top of hypocotyls of two independent transgenic lines, the nuclear/cytoplasmic ratio of GFP-PIF7 increased within 5 min of moving the plants into shade and continued to increase for 45 min (*Figure 1a,b*). The localization of GFP, which was used as the control, was not affected by shade (*Figure 1a,b*; *Figure 1—figure supplement 1a*).

Subcellular fractionation experiments using whole seedlings of 35S::*PIF7-Flash (9xMyc-6xHis-3xFLAG)* transgenic lines further demonstrated that PIF7-Flash was enriched in the nuclear fraction under shade conditions (*Figure 1c*, *Figure 1—figure supplement 1b*). Shade treatment resulted in an increase of PIF7 in the nucleus and a decrease of PIF7 in the non-nuclear fraction, indicating that the increased nuclear fraction of PIF7 was probably translocated from the cytoplasmic compartment.

### PIF7 interacts with 14-3-3 proteins

To identify potential PIF7-binding proteins, we conducted a yeast two-hybrid (Y2H) screen using PIF7 as bait. Interestingly, this study identified the phosphopeptide-binding protein 14-3-3 κ as a binding partner of PIF7. As we have shown, 14-3-3 λ and 14-3-3 κ can interact with PIF7 when co-expressed in the yeast system (*Figure 2a*). A bimolecular fluorescence complementation (BiFC) assay in *Nicotiana benthamiana* cells also supported the interaction between PIF7 and 14-3-3 λ/κ (*Figure 2b*). In fact, there are at least six 14-3-3 proteins (14-3-3 λ, κ, χ, γ, μ and ε) that can interact with PIF7 in Y2H and BiFC assays (*Figure 2a,b*; *Figure 2—figure supplement 1*).

The 14-3-3 proteins are well known to bind phosphopeptides. When GST (glutathione S-transferase)−14-3-3 fusion proteins were used to pull down the protein lysate from the *35S::PIF7-Flash* transgenic line grown under white-light and shade conditions, more PIF7 protein was enriched in the white-light-grown seedlings (*Figure 2c*). Moreover, a co-immunoprecipitation experiment further confirmed that 14-3-3 proteins are precipitated with PIF7 from white-light-grown transgenic seedlings (*Figure 2d*), probably because more PIF7 is phosphorylated in white-light-grown seedlings.

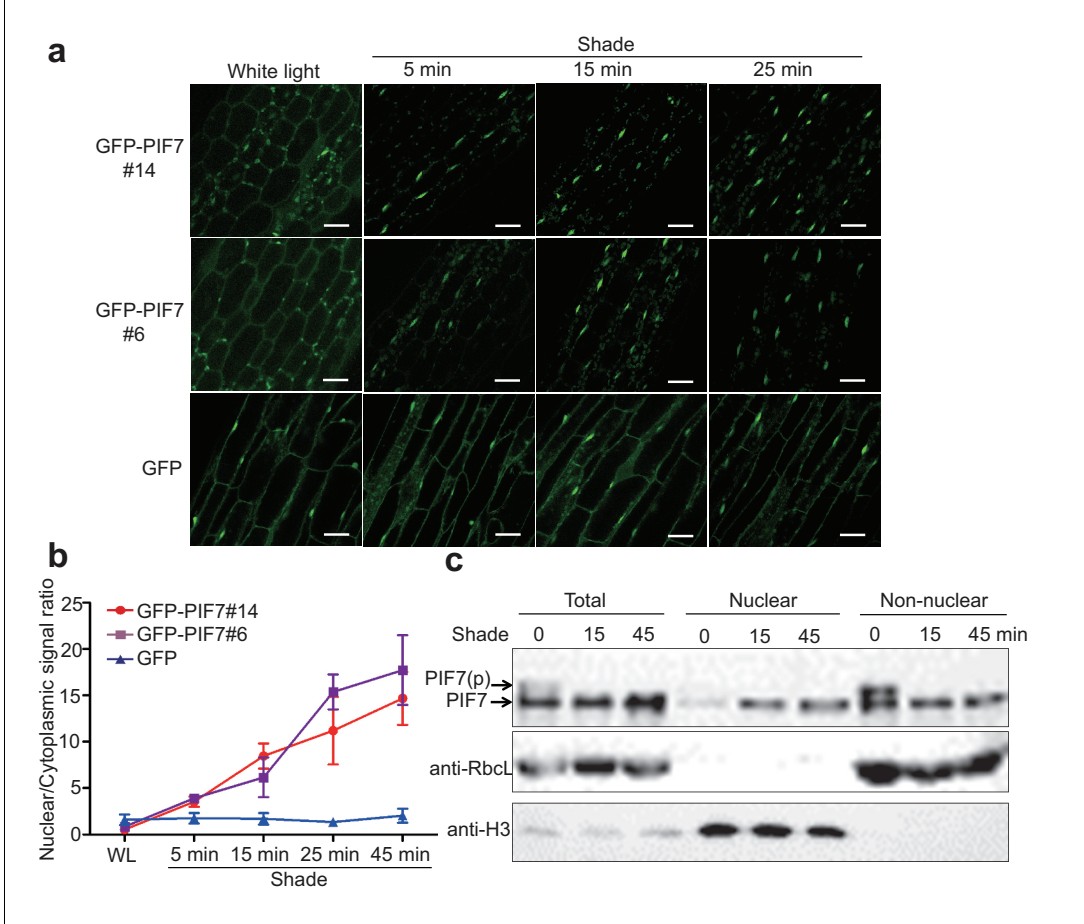

**Figure 1.** Shade induces the nuclear localization of PIF7. (a) Subcellular localization of GFP-PIF7 at the top of the hypocotyls of two independent transgenic seedlings grown under white light at different time points after transfer to shade. Transgenic *Arabidopsis* expressing GFP-PIF7 or GFP was grown on 1/2 MS medium under white light for 5 days. Seedlings were treated with shade for 5, 15, or 25 min, and images of the GFP signal were obtained using confocal microscopy. White scale bar represents 25 μm. (b) Kinetics of the shade-induced nuclear accumulation of GFP-PIF7. GFP-PIF7 or GFP seedlings were treated as in (a). ImageJ was used to quantify the fluorescence intensities. Ratios of the nuclear and cytoplasmic signal intensities were calculated from 10 cells for each treatment. Error bars represent standard deviations. (c) Shade induces the nuclear localization of dephosphorylated PIF7. Immunoblot of the PIF7-Flash proteins using anti-Myc antibody in the total, nuclear and non-nuclear fractions from white-light- and shade-treated seedlings. Histone H3 is a nuclear marker, and the RuBisCO large subunit (RbcL), a chloroplast protein, is a non-nuclear fraction marker.

DOI: https://doi.org/10.7554/eLife.31636.002

The following source data and figure supplement are available for figure 1:

**Source data 1.** Source files for the ratios of the nuclear and cytoplasmic signal intensities in *Figure 1b*.
DOI: https://doi.org/10.7554/eLife.31636.004

**Figure supplement 1.** Shade induces the nuclear localization of PIF7 in the hypocotyl and cotyledon.
DOI: https://doi.org/10.7554/eLife.31636.003

## Phosphorylation sites of PIF7 mediate its binding to 14-3-3 proteins

There are two types of specific 14-3-3 binding motifs in mammalian and plant systems, mode I, R/KXXpSX, and mode II, R/KXXXpSXP (where X = any amino acid, R = arginine, K = lysine, pS = phosphoserine and p=proline) (*Muslin et al., 1996*; *Muslin and Xing, 2000*; *Schoonheim et al., 2007*). Although an obvious interaction occurred between PIF7 and the 14-3-3 proteins, no typical mode I or mode II motifs can be identified in PIF7. We reasoned that non-canonical motifs, such as RXXS, might mediate the interaction between PIF7 and 14-3-3 proteins, as observed in PHOT1 (*Figure 3—figure supplement 1a*) (*Kinoshita et al., 2003*).

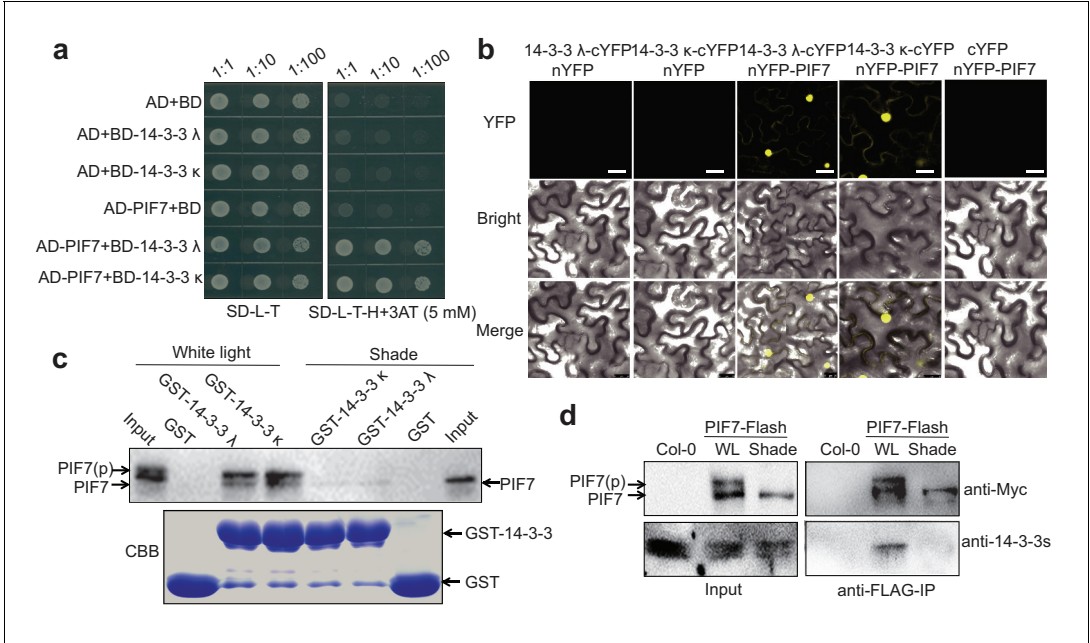

**Figure 2.** PIF7 interacts with 14-3-3 proteins. (**a**) PIF7 interacts with 14-3-3 λ and 14-3-3 κ in a yeast two-hybrid assay. Each yeast clone containing pGADT7 (AD) or pGADT7-PIF7 (AD-PIF7), together with pGBKT7 (BD), pGBKT7-14-3-3 λ (BD-14-3-3 λ) or pGBKT7-14-3-3 κ (BD-14-3-3 κ), was grown on transformation selection (SD-L-T) or interaction selection (SD-L-T-H+3AT) plates. Dilution of the inoculation is shown at the top of the picture. Yeast growth on SD-L-T-H+3AT indicates a positive protein–protein interaction. (**b**) Interaction between PIF7 and 14-3-3 λ or 14-3-3 κ as detected by BiFC. The C-terminal half of yellow fluorescent protein (YFP) was fused to 14-3-3 λ or 14-3-3 κ and the N-terminal half of YFP was fused to PIF7. The constructs were co-transformed into tobacco leaf cells, and fluorescence images were obtained by confocal microscopy. White scale bar represents 75 μm. (**c**) Interaction between PIF7 and 14-3-3 λ or 14-3-3 κ as detected by semi-in vivo pull-down assay. 14-3-3 λ and 14-3-3 κ fused to GST were expressed and purified from *Escherichia coli*. Protein extracts from plants that overexpressed PIF7-Flash, grown under white light conditions or after 1 hr of shade, were used for the pull-down assay. Immunoblots of the PIF7-Flash proteins used anti-Myc antibody. CBB: Coomassie Brilliant Blue stain. (**d**) Interaction between PIF7 and 14-3-3s as detected by co-immunoprecipitation. Anti-FLAG M2 agarose beads were used to precipitate PIF7-Flash from PIF7 overexpression plants grown under white light or after 1 hr of shade. Western blots using anti-Myc and anti-14-3-3s antibodies were performed as indicated in the 'Materials and methods'.

DOI: https://doi.org/10.7554/eLife.31636.005

The following figure supplement is available for figure 2:

**Figure supplement 1.** PIF7 can interact with 14-3-3 χ, γ, μ and ε proteins.

DOI: https://doi.org/10.7554/eLife.31636.006

In order to identify the phosphorylation sites of PIF7, we immunoprecipitated the PIF7 complex in 35S::*PIF7-Flash* transgenic plants and then performed a liquid chromatography-tandem mass spectrometry (LC-MS/MS) experiment. The results showed that PIF7 was phosphorylated at S139 and S141 in seedlings grown under while light and in seedlings treated with 5 min of shade, but not in seedlings treated with 1 hr of shade (*Figure 3—figure supplement 1b*). These phosphorylation sites constituted the putative 14-3-3 binding sequence (RSGSET). Because the LC-MS/MS experiment did not cover the entire protein, we also used the online software NetPhos 2.0 to predict the potential phosphorylation sites of PIF7, and the results showed the following high-score sites: S78, S80, S125, S139 and S141 (*Figure 3—figure supplement 1c*). On the basis of the 14-3-3-binding sequence in CDC25C (KTVSLC) (*Chan et al., 2011*), the sequence KDGSCS (75–80) of PIF7 may be another possible 14-3-3 binding motif (*Figure 3—figure supplement 1a*).

To determine whether the amino acids 138–143 (RSGSET) and 75–80 (KDGSCS) of PIF7 could mediate the binding of PIF7 to the 14-3-3 proteins, we mutated both S139 and S141 to alanine (to mimic the unphosphorylated state of these residues, resulting in PIF7[2A]: S139A S141A), deleted amino acids 138–141 (PIF7△), or mutated all five serine residues at amino acids 78, 80, 125, 139 and 141 (PIF7[5A]: S78A S80A S125A S139A S141A) of PIF7 to examine its interaction with 14-3-3

proteins. We also mutated S139 and S141 to aspartic acid residues to mimic phosphorylation (PIF7 [2D]: S139D S141D) (PIF7[5D]: S78D S80D S125D S139D S141D) (*Figure 3—figure supplement 1a*).

In yeast, PIF7(2A) showed decreased interaction with 14-3-3λ/κ proteins and PIF7(2D) displayed a level of interaction that was similar to that of wildtype PIF7 (*Figure 3a*). In tobacco leaves, the weak BiFC signals from PIF7(2A) and PIF7△ mainly occurred in the nucleus (*Figure 3b*; *Figure 3—figure supplement 2*), which is probably due to the nuclear localization of PIF7(2A) and PIF7△ or to an attenuated interaction with 14-3-3 λ/κ proteins. When all five serine residues were mutated (PIF7 [5A]), fluorescence was largely absent (*Figure 3b*, *Figure 3—figure supplement 2*), indicating that these serine residues are critical for binding to 14-3-3 proteins. However, no signal was observed from PIF7(2D) or PIF7(5D) in a BiFC assay (*Figure 3—figure supplement 3a*). Furthermore, PIF7(2D) extracted from a white-light-grown PIF7(2D)-Flash transgenic line was not able to pull down GST-14-3-3s (*Figure 3—figure supplement 3c*), probably because the interaction of the 14-3-3 protein with phosphorylated PIF7 cannot be mimicked by S-to-D substitution in some systems, as has also been

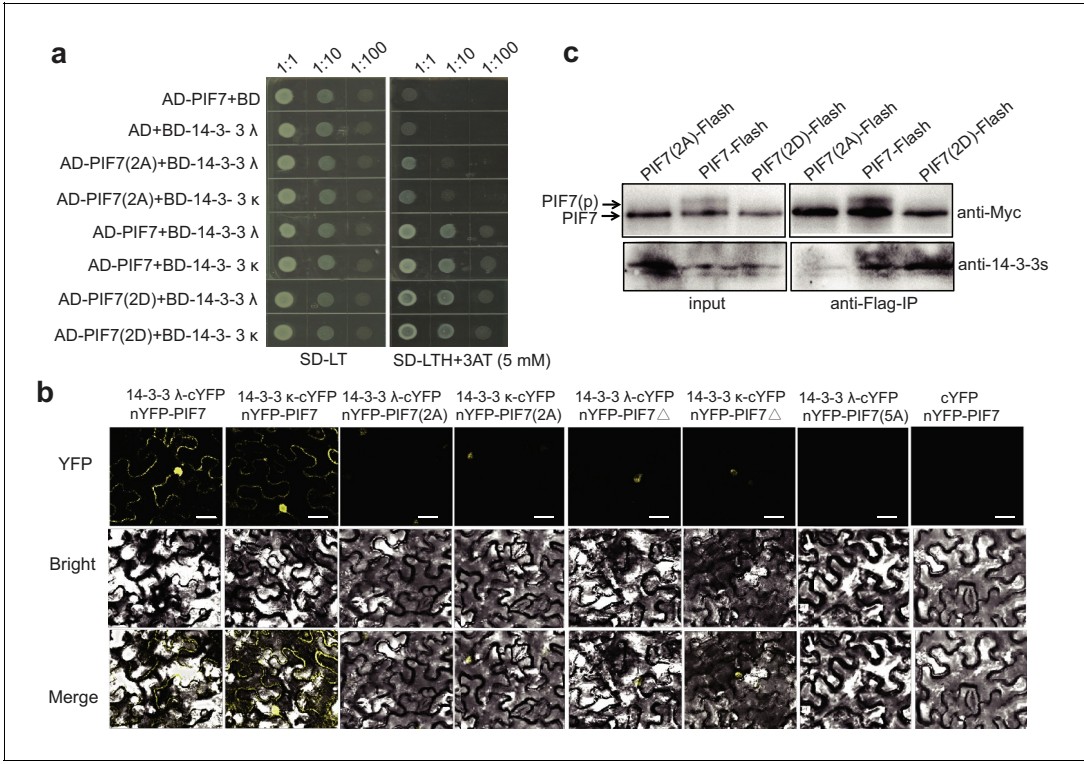

**Figure 3.** Interactions between PIF7 derivatives and 14-3-3 proteins. (**a**) Interactions between PIF7, PIF7(2A) or PIF7(2D) and 14-3-3 λ or 14-3-3 κ in a yeast two-hybrid assay. Each yeast clone containing the pGADT7 (AD), or AD-PIF7 (2A), or AD-PIF7 or AD-PIF7 (2D) together with pGBKT7 (BD) or BD-14-3-3 λ or BD-14-3-3 κ was grown on transformation selection (SD-L-T) or interaction selection (SD-L-T-H+3AT) plates. Dilution of the inoculation is shown at the top of the picture. Yeast growth on SD-L-T-H+3AT indicates a positive protein–protein interaction. (**b**) Interaction between PIF7 derivatives and 14-3-3 λ or 14-3-3 κ detected by BiFC. nYFP-PIF7, nYFP-PIF7(2A), nYFP-PIF7△ or nYFP-PIF7(5A) and 14-3-3 λ-cYFP or 14-3-3 κ-cYFP constructs were co-transformed into tobacco leaf cells. YFP fluorescence images were obtained using a confocal microscope. White scale bar represents 75 μm. (**c**) Interaction between PIF7, PIF7(2A) or PIF7(2D) and 14-3-3s as detected by co-immunoprecipitation. Anti-FLAG M2 agarose beads were used to precipitate PIF7-Flash, PIF7(2A)-Flash or PIF7(2D)-Flash from overexpression plants grown under white light. Western blots were performed using anti-Myc and anti-14-3-3s antibodies as indicated in the 'Materials and methods'.

DOI: https://doi.org/10.7554/eLife.31636.007

The following figure supplements are available for figure 3:

**Figure supplement 1.** Potential phosphorylation sites on PIF7.

DOI: https://doi.org/10.7554/eLife.31636.008

**Figure supplement 2.** Expression of 14-3-3 proteins and PIF7, and quantification of YFP signals in *Figure 3b*.

DOI: https://doi.org/10.7554/eLife.31636.009

**Figure supplement 3.** Interactions between PIF7, PIF7(2A) or PIF7(2D) and 14-3-3 λ or 14-3-3 κ in BiFC and in a pull-down assay.

DOI: https://doi.org/10.7554/eLife.31636.010

reported for other clients of 14-3-3 proteins (*de Chiara et al., 2009*; *Menon et al., 2012*). In a semi-in vivo pull-down assay, PIF7(2A)-Flash from the white-light-grown 35S::*PIF7(2A)-Flash* transgenic line showed reduced levels of binding with GST-14-3-3 fusion proteins (*Figure 3—figure supplement 3b*). Finally, in an in vivo Co-IP assay, more 14-3-3 proteins were co-immunoprecipitated with PIF7(2D) than with wild type PIF7 or PIF7(2A) (*Figure 3c*), suggesting that the phosphorylation sites of PIF7 mediated its binding to 14-3-3 proteins.

## Phosphorylation sites of PIF7 are important for its localization and function

To investigate the roles of the phosphorylation sites on the cellular localization of PIF7, we generated 35S::*GFP-PIF7* derivatives in which the serine residues were mutated to alanine residues (PIF7 [2A] and PIF7[5A] or to aspartic acid residues (PIF7[2D] and PIF7[5D]). When over-expressed in tobacco leaf cells, the wildtype GFP-PIF7 localized in both the cytoplasm and the nucleus. The GFP-PIF7(2A), GFP-PIF7(5A) and GFP-PIF7△ mutants exhibited strong nuclear signals, whereas the GFP-PIF7(2D) and GFP-PIF7(5D) mutants showed stronger cytoplasmic signals than did GFP-PIF7 (*Figure 4a*; *Figure 4—figure supplement 1*). To further examine the effect of the phosphorylation sites on PIF7 localization in *Arabidopsis*, we generated transgenic plants expressing GFP-PIF7△ and GFP-PIF7(5A). Consistent with the findings in tobacco, GFP-PIF7△ and GFP-PIF7(5A) displayed stronger nuclear signals than did wildtype GFP-PIF7 (*Figure 4b*). The shade treatment caused substantial translocation of the wildtype GFP-PIF7, whereas its effects on GFP-PIF7△ and GFP-PIF7(5A) were minimal (*Figure 4b*).

To determine the effect of the phosphorylation sites of PIF7 on the shade response in vivo, PIF7 with phosphorylation site mutations (35S::*PIF7[2A]- Flash*, 35S::*PIF7[2D]-Flash*) and wildtype (35S::*PIF7-Flash*) were expressed in the *pif7-1* background, and the number of complementary transgenic lines was recorded on the basis of hypocotyl length under shade. A substantial proportion of the first generation (T1) of transgenic plants transformed with PIF7 (13/118: 11%) and PIF7(2A) (47/212: 22%) showed normal or enhanced shade responses. By contrast, all of the transgenic plants transformed with 35S:: *PIF7(2D)-Flash* (83 T1 lines) showed a shade-defective phenotype similar to that of *pif7-1*. To further confirm the phenotype, we obtained the third-generations of the 35S::*PIF7(2A)-Flash*, 35S::*PIF7(2D)-Flash* and 35S::*PIF7-Flash* transgenic lines and confirmed that their expression levels were similar to those of the equivalent T1 lines. As we have shown, the phosphorylated form of PIF7 was decreased in the PIF7(2A) transgenic lines (*Figure 4c*). In a subcellular fractionation experiment, PIF7(2A) was present in the nuclear fraction, but almost undetectable in the cytosol under both white light and shade (*Figure 4d*). By contrast, PIF7(2D) was mainly present in the cytoplasmic fraction (*Figure 4e*). Furthermore, when grown under white light, the *PIF7* and *PIF7(2A)* transgenic plants showed greatly increased expression of the PIF7 downstream genes *IAA19* and *YUCCA8* (*Figure 4f*), resulting in longer hypocotyls (*Figure 4g*). By contrast, overexpression of PIF7(2D) in the *pif7-1* background resulted in reduced shade-induction of both gene expression (*Figure 4f*) and hypocotyl elongation (*Figure 4g*). Notably, the effects of PIF7(2A) overexpression were greater than those of PIF7 overexpression, indicating the nuclear localization is critical for the function of PIF7. It is also noteworthy that the shade-induced effects on gene expression and hypocotyl elongation were not totally abolished in the *PIF7(2A)* transgenic lines (*Figure 4f,g*), implying that other phosphorylation sites or other mechanisms that regulate PIF7 may exist. For example, shade may regulate additional essential factors that interact with PIF7.

## 14-3-3 proteins regulate the localization and dephosphorylation of PIF7

As PIF7 translocates to the nucleus under shade conditions and is able to bind 14-3-3 proteins, we hypothesized that 14-3-3 proteins sequester phosphorylated PIF7 in the cytoplasm.

Previous studies have shown that the interactions of 14-3-3 proteins with their client proteins can be disrupted by the R18 peptide (*Wang et al., 1999*). We therefore took advantage of this peptide to discover that the shade-induced nuclear localization of GFP-PIF7 (*Figure 5a,b*) and de-phosphorylation of PIF7-Flash (*Figure 5c*; *Figure 5—figure supplement 1*) were accelerated after treatment with R18, but not after treatment with R18(Lys) (a non-functional control) (*Figure 5—figure supplement 2*). In addition, we crossed 35S::*GFP-PIF7* and 35S::*PIF7-Flash* transgenic lines with the double mutant *14-3-3 λκ*, in which the expression of 14-3-3s was reduced (*Zhou et al., 2014*). When

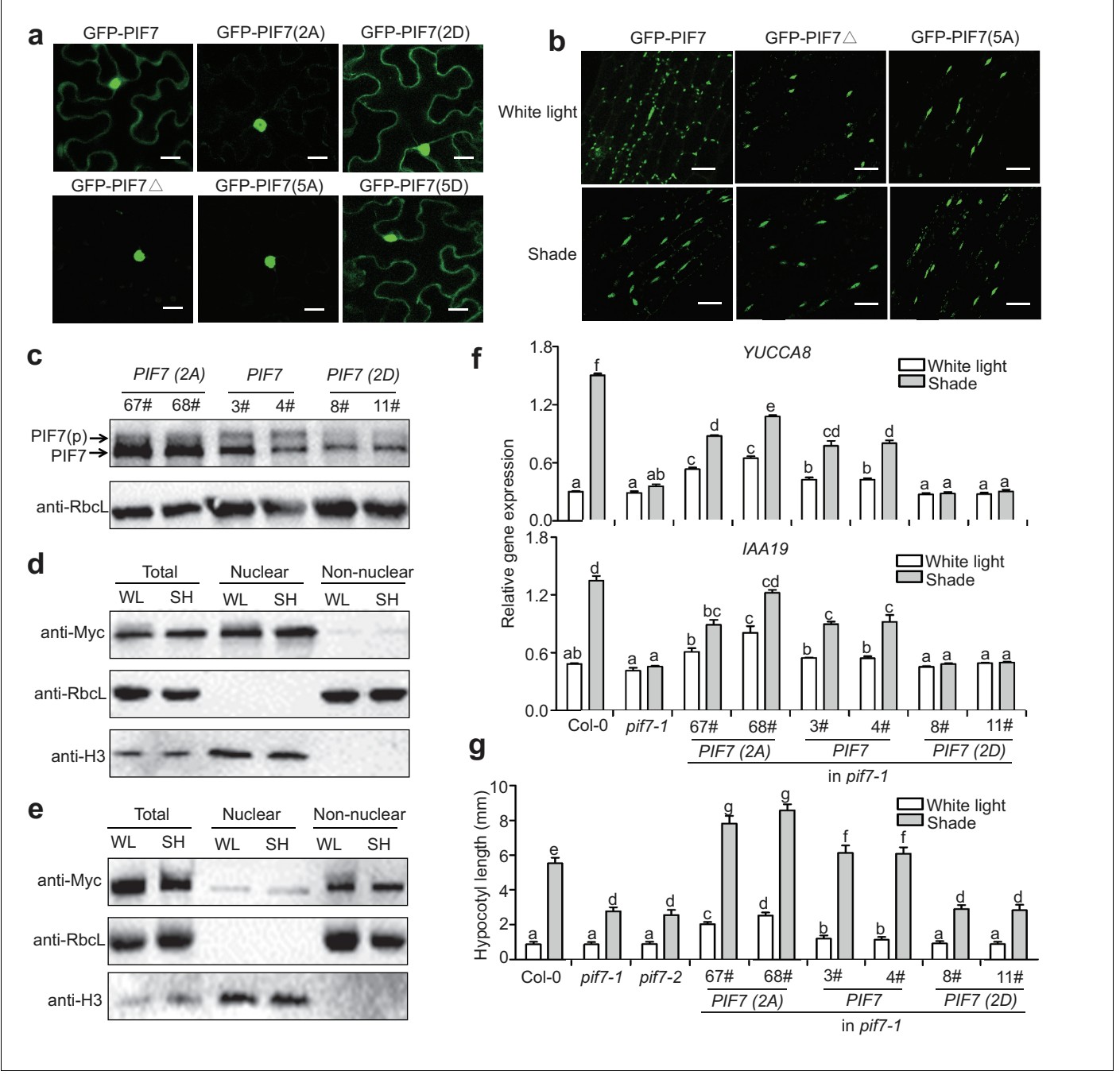

**Figure 4.** Phosphorylation sites of PIF7 are important for the localization and function of this protein. (a) Subcellular localization of GFP-PIF7, GFP-PIF7 (2A), GFP-PIF7 (2D), GFP-PIF7△, GFP-PIF7(5A) and GFP-PIF7 (5D) in tobacco cells. White scale bar represents 25 μm. (b) Subcellular localization of the GFP-PIF7, GFP-PIF7△ and GFP-PIF7(5A) proteins at the top of the hypocotyl in transgenic *Arabidopsis* plants treated with white light or after 1 hr in shade. White scale bar represents 25 μm. (c) Expression of PIF7 in white-light-grown 35S::*PIF7(2A)-Flash*, 35S::*PIF7-Flash* and 35S::*PIF7(2D)-Flash* transgenic lines as determined using anti-Myc antibody. RbcL is used as the loading control. (d) Subcellular fractionation experiments using 35S::*PIF7 (2A)-Flash* transgenic lines. Immunoblot of the PIF7(2A)-Flash proteins using anti-Myc antibody in the total, nuclear and non-nuclear fractions from white-light- and shade-treated transgenic seedlings. Histone H3 is a nuclear maker and RbcL is a non-nuclear fraction marker. (e) Subcellular fractionation experiments using 35S::*PIF7(2D)-Flash* transgenic lines. Immunoblot of the PIF7(2D)-Flash proteins using anti-Myc antibody in the total, nuclear and non-nuclear fractions from white-light- and shade-treated transgenic seedlings. Histone H3 is a nuclear marker and RbcL is a non-nuclear fraction marker. (f) Expression levels of *IAA19* and *YUCCA8* in Col-0, *pif7-1* and transgenic lines harboring PIF7-Flash, PIF7(2A)-Flash and PIF7(2D)-Flash in the *pif7-1* background. Mean ± SE from three independent biological replicates, after normalization to the internal control *AT2G39960*, are shown. Bars marked with different letters denote significant differences (p<0.05) in the mean expression levels. (g) Quantification of the hypocotyl lengths of transgenic lines harboring PIF7-Flash, PIF7(2A)-Flash and PIF7(2D)-Flash in the *pif7-1* background. Seedlings were grown under white light for 4 days

*Figure 4 continued on next page*

*Figure 4 continued*

and maintained in white light or transferred to shade for the next 5 days, before hypocotyl lengths were measured. More than 20 seedlings were measured for each line. Bars marked with different letters denote significant differences (p<0.05) in the mean hypocotyl lengths.

DOI: https://doi.org/10.7554/eLife.31636.011

The following figure supplement is available for figure 4:

**Figure supplement 1.** Subcellular localization of GFP-PIF7, GFP-PIF7(2A), GFP-PIF7(2D), GFP-PIF7△, GFP-PIF7(5A) and GFP-PIF7 (5D) in tobacco cells.

DOI: https://doi.org/10.7554/eLife.31636.012

compared with wildtype background, more GFP-PIF7 accumulated in the nucleus of the double mutant after 2 min of shade treatment (*Figure 5d,e*) and the phosphorylated PIF7 disappeared faster in the double mutant (*Figure 5f*; *Figure 5—figure supplement 2*), suggesting that 14-3-3 proteins mediate the cytoplasmic retention of phosphorylated PIF7 during the transition from white light to shade.

## 14-3-3 proteins are negative regulators of the shade response

Consistently, treatment with R18 significantly promoted shade-induced hypocotyl elongation (*Figure 6a*) and shade-induced gene (*IAA19* and *YUCCA8*) expression (*Figure 6b*). However, this promotion is blunted in R18-treated *pif7-1*. We also determined whether a loss of 14-3-3 λ or 14-3-3 κ function would affect shade-induced hypocotyl elongation in *Arabidopsis*. Single mutants for each gene and the double mutant displayed enhanced shade responses, whereas the 14-3-3 λ transgenic lines (14-3-3 λ OE and *35S::FLAG-HA-14-3-3 λ*) showed a reduced shade response (*Figure 6c*, *Figure 6—figure supplement 1*), as measured by hypocotyl elongation and levels of expression of *IAA19* and *YUCCA8* (*Figure 6d*). Moreover, with R18-treated *pif7-1*, the hypocotyl length of the double mutant of *pif7-1* and *14-3-3 λ−2* was consistently more like that in *pif7-1* (*Figure 6c*), suggesting that the function of 14-3-3 proteins is mediated by PIF7. Overall, the phenotype and gene expression analysis demonstrated that 14-3-3 λ and 14-3-3 κ negatively regulate the shade response through PIF7.

## Discussion

In the current study, we improved the model of shade signal transduction by demonstrating a shade-sensitive subcellular localization of PIF7, which is conferred by interactions with 14-3-3 proteins (*Figure 7*). Under white-light conditions, phosphorylation of PIF7 results in the cytoplasmic location of this protein and enables its binding to 14-3-3 proteins. When plants sense shade conditions, unknown phosphatases remove the phosphorylation of PIF7. De-phosphorylated PIF7 does not interact with 14-3-3 proteins and translocates to the nucleus, where it promotes the expression of downstream genes, leading to shade-induced phenotypic changes. 14-3-3 proteins retain phosphorylated PIF7 in the cytoplasm to regulate shade-induced hypocotyl elongation negatively.

Current and previous studies have provided several lines of evidence to strongly support the importance of the phosphorylation of PIFs for the transcriptional activity of these proteins. Several phosphorylation sites have been identified in PIF1/3/4 (*Bu et al., 2011*; *Ni et al., 2013*; *Bernardo-García et al., 2014*). These phosphorylation events lead to apparent ubiquitylation and degradation through the ubiquitin-proteasome system (*Al-Sady et al., 2006*). In sustained light, PIF1/3/4/5 are maintained at a relatively low steady-state level. After subsequent exposure to shade light, new protein synthesis is required for the accumulation of these PIF proteins (*Lorrain et al., 2008*). PIF7, however, is a light-stable bHLH factor (*Leivar et al., 2008*; *Li et al., 2012*).

In the current study, at least two phosphorylation sites (S139 and S141) in PIF7 are found to be critical for its activity and for hypocotyl elongation. These sites also mediate the binding of PIF7 with 14-3-3 λ and κ. Shade-induced nuclear accumulation of PIF7 and the constitutive nuclear localization of PIF7(2A) mutants in transgenic plants suggest that the interaction of 14-3-3 proteins and PIF7 is involved in determining the subcellular distribution of PIF7. Although there was a lack of interaction in the BiFC and pull-down assays, S-to-D substitution does enhance the interaction between 14-3-3 proteins and PIF7 in vivo. Moreover, PIF7(2D)-Flash was mainly localized in cytoplasm and was unable to complement the shade-defective gene expression and phenotype of *pif7-1*, which

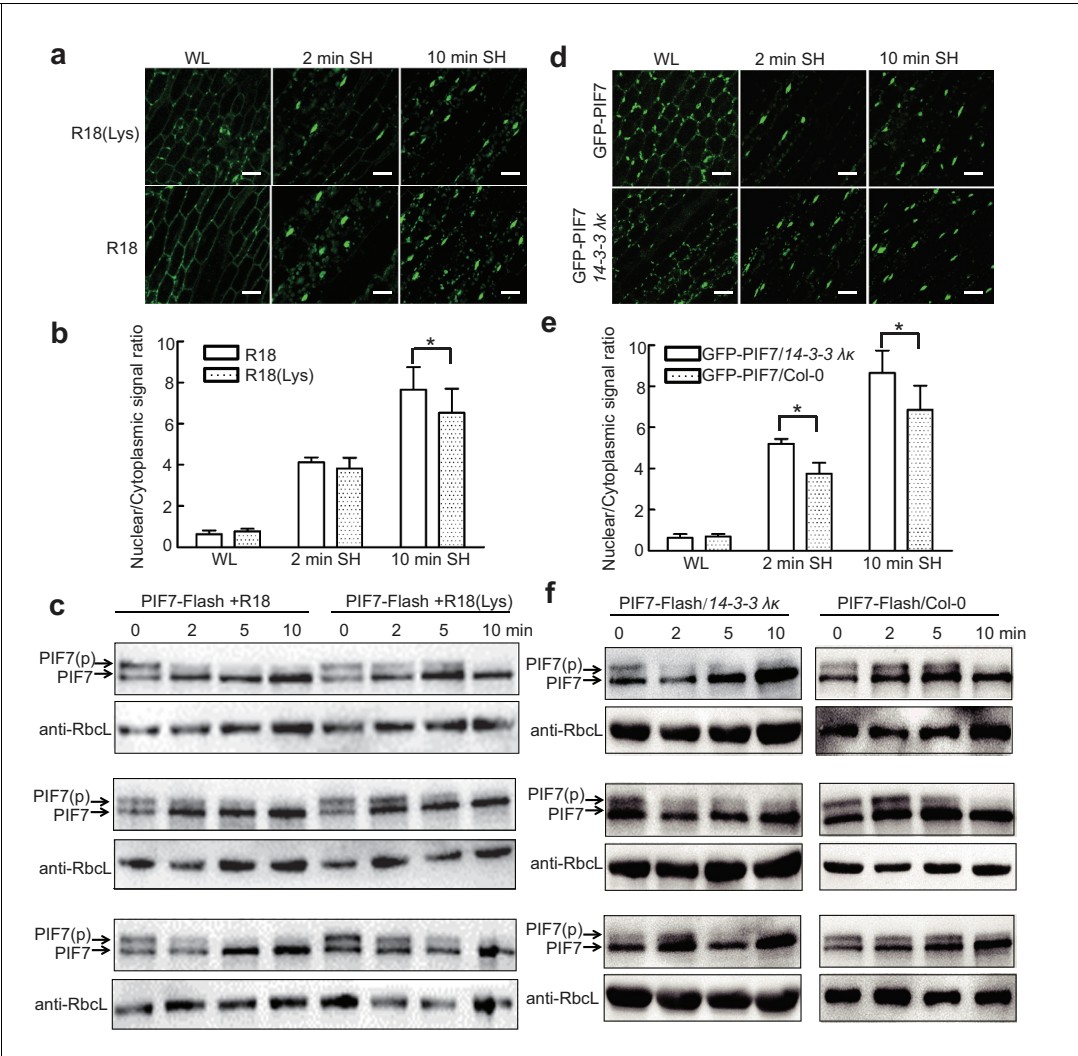

**Figure 5.** 14-3-3 proteins delay shade-induced nuclear translocation and dephosphorylation of PIF7. (a) The effect of R18 on the shade-induced nuclear localization of GFP-PIF7. GFP-PIF7 #14 transgenic plants grown under white light were treated with R18 or R18(Lys) for 3 hr after 10 min of vacuum, and were transferred to shade for indicated periods. White scale bar represents 25 μM. (b) Quantification of the shade-induced nuclear accumulation of GFP-PIF7. ImageJ was used to quantify the fluorescence intensities. Ratios of the nuclear and cytoplasmic signal intensities were calculated from 10 cells for each treatment. Error bars represent standard deviations. Significant differences between two treatments are shown as asterisks. *p<0.05 by Student's t-test. (c) The effect of R18 on the shade-induced dephosphorylation of PIF7-Flash. Five-day-old 35S::*PIF7-Flash* transgenic seedlings were treated with R18 or R18(Lys) for 3 hr after 10 min of vacuum under white light, and transferred to shade for the indicated periods. Data for three biological replicates are presented. The level of PIF7-Flash was detected using anti-Myc antibody. RbcL was used as the loading control. (d) Shade-induced nuclear localization of GFP-PIF7 in *14-3-3 λκ* (Salk-075219CxSalk-071097). A GFP-PIF7 #14 transgenic plant was crossed with *14-3-3 λκ*. Five-day-old white-light-grown GFP-PIF7/*14-3-3 λκ* and GFP-PIF7/Col-0 seedlings were transferred to shade for the indicated periods. White scale bar represents 25 μM. (e) Quantification of the shade-induced nuclear accumulation of GFP-PIF7. ImageJ was used to quantify the fluorescence intensities. Ratios of the nuclear and cytoplasmic signal intensities were calculated from 10 cells for each treatment. Error bars represent standard deviations. Significant differences between*14-3-3 λκ* and Col-0 background are shown as asterisks. *p<0.05 by Student's t-test. (f) Shade-induced dephosphorylation of PIF7-Flash in *14-3-3 λκ*. A 35S::*PIF7-Flash* transgenic plant was crossed with *14-3-3 λκ*. Five-day-old white-light-grown PIF7-Flash/*14-3-3 λκ* and PIF7-Flash/Col-0 seedlings were transferred to shade for the indicated periods. Three biological replicates were presented. The level of PIF7-Flash was detected using anti-Myc antibody. RbcL was used as the loading control.

DOI: https://doi.org/10.7554/eLife.31636.013

The following source data and figure supplements are available for figure 5:

**Source data 1.** Source files for the ratios of the nuclear and cytoplasmic signal intensities in *Figure 5b* and *Figure 5e*.

DOI: https://doi.org/10.7554/eLife.31636.016

**Source data 2.** Source files for the ratios of phosphorylated PIF7(PIF7[p]) relative to total PIF7 proteins in *Figure 5—figure supplement 1*.

DOI: https://doi.org/10.7554/eLife.31636.017

*Figure 5 continued on next page*

*Figure 5 continued*

**Figure supplement 1.** Quantification of phosphorylated PIF7(PIF7[p]) relative to total PIF7 proteins in *Figure 5c and f*.
DOI: https://doi.org/10.7554/eLife.31636.014
**Figure supplement 2.** R18(Lys) is a non-functional peptide control for R18.
DOI: https://doi.org/10.7554/eLife.31636.015

functionally reflects the phosphorylation defects in the in vivo system. The phosphorylation state of PIF7 determines its localization and function, and also affects its ability to bind to 14-3-3 proteins.

14-3-3 proteins have been reported to regulate transcription factors by sequestering them in the cytoplasm; for example, BZR1 and RSG are regulated by the binding of 14-3-3 λ, ω or μ (*Igarashi et al., 2001*; *Gampala et al., 2007*). Although functional redundancy and different combinations of 14-3-3 isoforms bring difficulties in clarifying the specific roles of 14-3-3 proteins (*Jaspert et al., 2011*), the involvement of 14-3-3 proteins in light signaling has been illustrated by the elongated hypocotyls (relative to those of Col-0 seedlings) of 14-3-3 κ, ν and χ mutants grown in red light (*Mayfield et al., 2007*; *Adams et al., 2014*). In our work, there was no significant effect of R18 treatment and 14-3-3 mutations on PIF7's localization and phosphorylation state under white light, which could be due to the strong light radiance, the inhibitory potency of R18 or the redundancy of the 13 14-3-3 proteins in *Arabidopsis*. By contrast, 14-3-3s significantly delay the shade-induced translocation and de-phosphorylation of PIF7 (*Figure 5*), and consequently enhance shade-induced hypocotyl elongation which is dependent on PIF7 (*Figure 6*). The weak shade phenotype might be caused by functional redundancy of 14-3-3 proteins. It is also possible that a compensatory increase in other isoforms occurs in *14-3-3 λκ*. It is possible that 14-3-3 proteins sequester phosphorylated PIF7 in cytoplasm by protecting the PIF7 proteins from phosphatases during the transition from white light to shade.

PIF7 is a major controller for shade-induced hypocotyl elongation, as demonstrated by the severe shade-defective phenotype of *pif7* mutants (*Li et al., 2012*). It is known that *Arabidopsis* grows rapidly in response to the shade stimulus, with an induction of *PIL1* transcript levels detectable after only 8 min of low R/FR and growth measurable after just 30 or 45 min of exposure to shade (*Salter et al., 2003*; *Cole et al., 2011*). A quick shade regulatory mechanism is required to achieve this rapid response. Phosphorylation-dependent translocation of PIF7 is such a quick mechanism that can give rise to efficient photomorphogenesis. Moreover, several negative regulators of PIF7 have been shown to reduce the transcriptional activity of light-responsive genes and to prevent exaggerated shade responses (*Hornitschek et al., 2009*; *Galstyan et al., 2011*; *Hao et al., 2012*; *Li et al., 2014*). In our current study, the binding of 14-3-3 proteins delays the de-phosphorylation and nuclear import of PIF7 in response to shading, forming another layer of regulation to determine the appropriate SAS.

A conserved 14-3-3-binding motif has been identified in PIF3 (RNP**S**PP), and PIF3 has been found to be a 14-3-3 interaction partner in an affinity-purification assay using His-tagged 14-3-3-coated beads (*Adams et al., 2014*). However, the disruption of putative phosphorylation sites on the 14-3-3-binding motifs of PIF3 did not prevent 14-3-3 from binding to PIF3 or disturb the nuclear localization of PIF3 (*Adams et al., 2014*). Although the 14-3-3-binding sites of PIF7 are not typical of those of the other PIFs, the functional analysis of mutations of these sites (from Ser to Ala) illustrated their critical roles in PIF7 function.

The expression level and localization of 14-3-3 λ protein remain stable after shading is introduced (*Figure 6—figure supplement 1*), suggesting that 14-3-3 proteins probably exert their cargo function constantly. The activity of PIF7 is mostly determined by its phosphorylation status, which is controlled by a kinase and a phosphatase that are light-dependent. To date, CK2 (*Bu et al., 2011*), PPKs (*Ni et al., 2017*) and BIN2 (*Bernardo-García et al., 2014*) have been reported to be the kinases of PIF1, PIF3 and PIF4, and TOPP4 has been reported to be the phosphatase of PIF5 (*Yue et al., 2016*); however, no specific kinase or phosphatase of PIF7 has yet been identified. The shade-induced localization response of PIF7 varied in the different tissues (*Figure 1—figure supplement 1*), implying that the phosphorylation of PIF7 is probably regulated by upstream signals that have tissue and/or developmental specificity. One important goal for the future is to identify these upstream kinases, as well as the phosphatase(s) responsible for the de-phosphorylation of Ser 139 and 140, and hence for the disassociation of 14-3-3 proteins in response to shading.

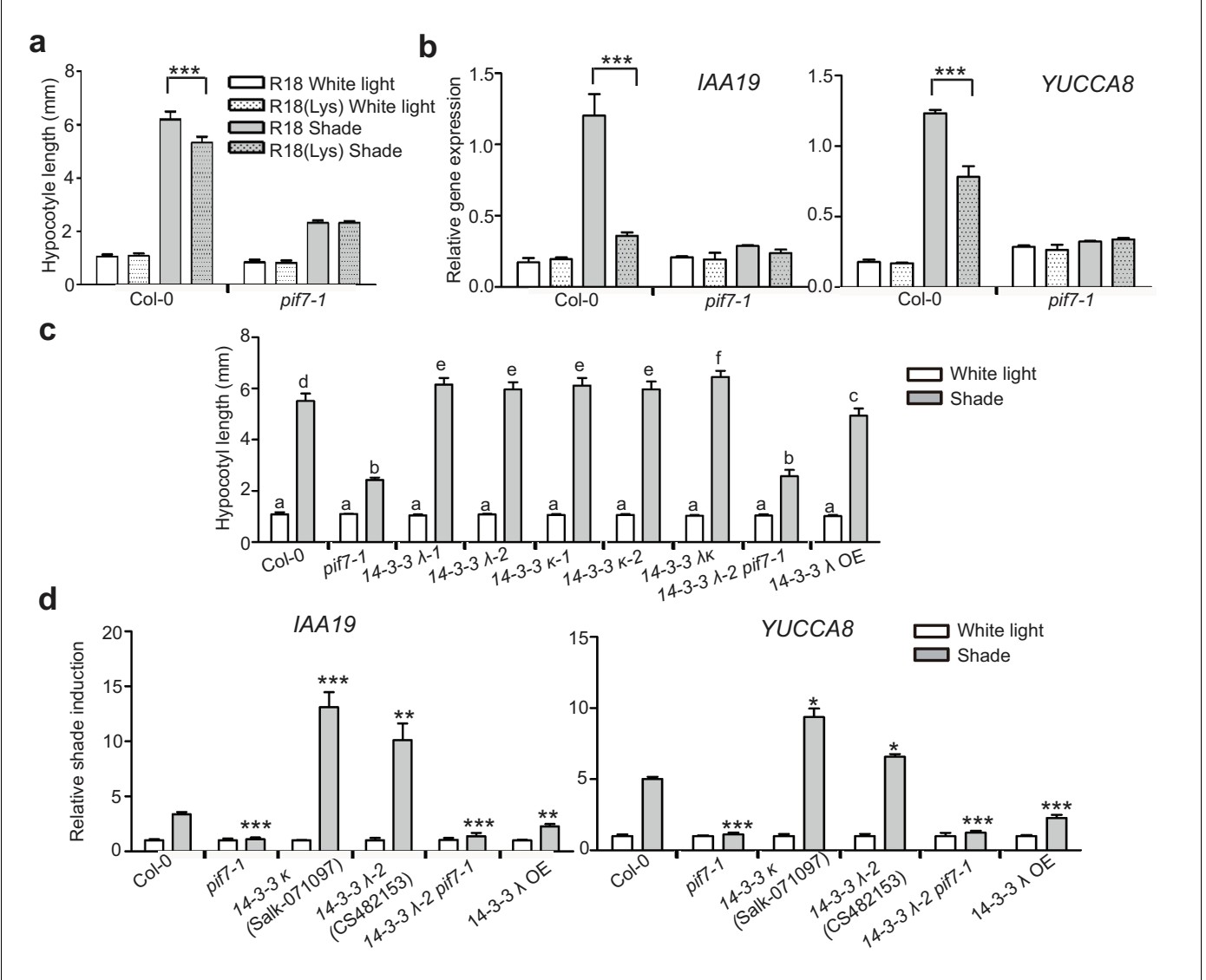

**Figure 6.** 14-3-3 proteins negatively regulate shade-induced hypocotyl elongation and gene expression. (a) Quantification of the hypocotyl length of Col-0 seedlings grown in plates containing 200 µg/ml R18 or R18(Lys) peptide under white light or shade conditions. Seedlings were grown under white light for 4 days and maintained in white light or transferred to shade for next 5 days before the measurement of hypocotyl length. More than 20 seedlings were measured. Significant differences between two treatments are shown as asterisks. ***p<0.001 by Student's t-test. (b) *IAA19* and *YUCCA8* expression level in the Col-0 and *pif7-1* seedlings treated with R18 and R18(Lys) under white light or shade. Seedlings were grown with 1/2 MS medium containing 200 µg/ml R18 or R18(Lys) under white light for 5 days. Then, the seedlings were kept in white light or transferred to shade for 1 hr. Mean ± SE from three independent biological replicates, after normalization to the internal control AT2G39960, are shown. Significant differences between two treatments are indicated by asterisks. ***p<0.001, by Student's t-test. (c) Quantification of the hypocotyl length of Col-0, *pif7-1*, *14-3-3* mutants and the overexpression line grown under white light or shade. More than 20 seedlings were measured. Bars marked with different letters denote significant differences (p<0.05) of the means of hypocotyl length. (d) Shade induction of *IAA19* and *YUCCA8* in Col-0, *pif7-1*, *14-3-3* mutants and 14-3-3 λ OE. The seedlings were grown under white light for 5 days. Then, the seedlings were kept in white light or transferred to shade for 1 hr. The expression levels were normalized to a reference gene (*AT2G39960*) and then normalized to the expression under white light condition. The relative shade inductions were shown. Significant differences between mutants and Col-0 are indicated by asterisks. *p<0.05, **p<0.01,***p<0.001 by Student's t-test.

DOI: https://doi.org/10.7554/eLife.31636.018

The following figure supplement is available for figure 6:

**Figure supplement 1.** Effects of shade on the mRNA level, protein level and localization of 14-3-3 λ.

DOI: https://doi.org/10.7554/eLife.31636.019

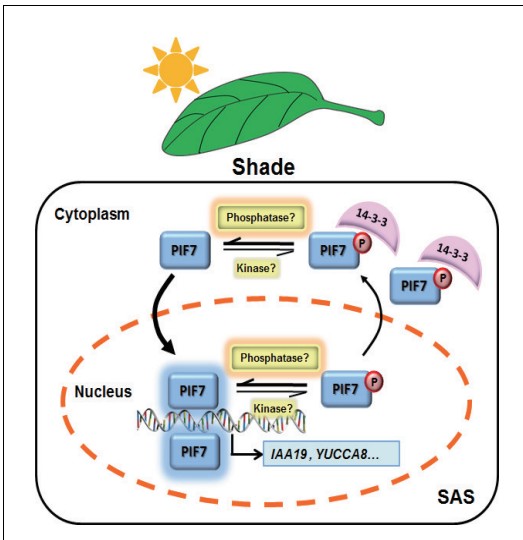

**Figure 7.** A molecular model illustrating the role of 14-3-3 proteins in PIF7-mediated SAS. In response to shade light, de-phosphorylated PIF7 accumulates in the nucleus. 14-3-3 proteins retain phosphorylated PIF7 in the cytoplasm and hence regulate shade-induced hypocotyl elongation negatively.

DOI: https://doi.org/10.7554/eLife.31636.020

Our findings offer novel insights into the mechanism through which a key transcription factor is activated by light. We already knew that a high R/FR light ration promotes the nuclear import of phyB, which positively regulates photomorphogenesis. Here, we propose that shade promotes the nuclear import of PIF7, which promotes SAS. Subcellular translocations contribute to the antagonistic action of phyB and PIF7. More detailed cooperation on light-mediated development will be necessary in the future.

## Materials and methods

### Plant materials and growth conditions

All of the *Arabidopsis thaliana* plants used in this study were of the Columbia-0 ecotype. The mutants of the 14-3-3 λ T-DNA lines (Salk_075219C and CS482153) and the 14-3-3 κ T-DNA lines (Salk_148929C and Salk_071097) were obtained from ABRC. The 14-3-3 λ and 14-3-3 κ double mutant (Salk_071097X Salk_075219C), the 35S::*FLAG-HA-14-3-3* λ transgenic plants (14-3-3 λ OE) (*Zhou et al., 2014*) and the 35S::*PIF7-Flash* plants (*Li et al., 2012*) have been described previously. For the phenotypic analysis, seeds were germinated on 1/2 Murashige and Skoog (MS) medium (Duchefa Biochemie, Haarlem, The Netherlands) plates with 1% agar (Sangon, Shanghai, China) and without sucrose. After stratification, the plates were incubated in growth chambers under continuous white light for 4 days at 22°C, and then the plates were either left in white light or transferred to canopy shade for 5 days before hypocotyl measurements were performed. HR-350 (HiPoint, Taiwan) was used to measure the light conditions. The shade conditions were as described for previous studies (*Tao et al., 2008*; *Li et al., 2012*). *Nicotiana benthamiana* plants were grown at 26°C under 16-hr-light long-day conditions.

### Identification of PIF7-interacting proteins by LC-MS/MS

The different light-treated 35S::*PIF7-Flash* seedlings were ground to a fine powder in liquid nitrogen and solubilized with lysis buffer (50 mM Tris-HCl [pH 8.0], 150 mM NaCl, 2 mM DTT, 1% NP40, 10% glycerol and protease inhibitor cocktail) (Roche, USA). The extracts were incubated at 4°C for 1 hr on a rotating wheel, and the insoluble material was removed by centrifugation at 20,000 x g at 4 °C for 15 min three times until the supernatant was clear. The supernatant was incubated with a pre-washed anti-FLAG M2 agarose gel (Cat# A2220 RRID: AB_10063035, Sigma-Aldrich, USA) at 4 °C for 3 hr in the rotating wheel. The beads were recovered by centrifugation at 800 rpm at 4 °C for 2–5 min. After six washes with lysis buffer, SDS-loading buffer was added to the pellet fraction. The samples were boiled for 5 min, centrifuged at maximum speed for 15 min, and then loaded onto an SDS-PAGE gel. After running the gel halfway, the gel was cut into 1 mm$^3$ cubes and sent for MS analysis. The trypsin-digested peptides were concentrated and analyzed using a Finniqan LTQ mass spectrometer (Thermoquest, San Jose, USA) coupled with a surveyor HPLC system.

### Isolation of the nuclear fraction

For nuclear fractionation, the different light-treated seedlings were ground to a fine powder and lysed with a buffer (20 mM Tris-HCl [pH 7.5], 20 mM KCl, 2 mM EDTA, 2.5 mM MgCl$_2$, 250 mM sucrose, 5 mM DTT, 25% glycerol and protease inhibitor cocktail) and filtered through Miracloth (Calbiochem, San Diego, USA). The obtained solution was centrifuged at 1500 x g for 10 min. The

pellet was washed five times with a re-suspension buffer (NRBT: 20 mM Tris-HCl [pH 7.5], 2.5 mM MgCl$_2$, 0.2% Triton X-100, 25% glycerol and protease inhibitor cocktail). After washing, the pellet was suspended in NRB2 (20 mM Tris-HCl [pH 7.5], 250 mM sucrose, 10 mM MgCl$_2$, 0.5% Triton X-100, 5 mM β-mercaptoethanol and protease inhibitor cocktail) and slowly added on top of the same volume of NRB3 (20 mM Tris-HCl [pH 7.5], 1.7 M sucrose, 10 mM MgCl$_2$, 0.5% Triton X-100, 5 mM β-mercaptoethanol and protease inhibitor cocktail). After centrifugation at 16,000 x g for 45 min at 4°C, the pellet was suspended in lysis buffer as the nuclear fraction.

## Confocal microscopy and quantitation of the fluorescent protein signal

The fluorescence images of GFP and YFP expression were obtained with a Leica confocal microscope (Leica SP8) at 488 and 514 nm. GFP-PIF7 transgenic lines were grown in white light and then transferred to the shade for 0, 5 15, 25, and 45 min. The fluorescence at each time point was recorded using a 40 × 1.3 objective lens. ImageJ (http://rsb.info.nih.gov/ij/) was used to quantify the fluorescence intensities. The images were converted to an 8-bit format, and the fluorescence intensity was integrated from all pixels in the selected area. To measure the ratio between the nuclear and cytoplasmic signals for each cell, the entire cellular and nuclear area was selected for quantification of fluorescence intensity. The cytoplasmic intensity was calculated by subtracting the value for the nuclear area from that for the whole cell. The ratio between the nuclear and cytoplasmic signals was calculated for 10 cells, and three repeated measurements were performed in each condition.

## Yeast two-hybrid screens and assay

A Matchmaker Gold Yeast Two-Hybrid system was used. The CDS of *PIF7* was cloned into a pGBKT7 vector and used as bait to identify interacting proteins from a cDNA library, which was prepared by Oebiotech (China) with RNA from Col-0 seedlings grown under white-light conditions for 4 days. The cDNA synthesized from this material was cloned into the bait vector pGADT7. The interactions were tested on SD medium without Leu, Trp, and His but with 5 mM 3-amino-1,2,4-triazole (3AT, Sigma), using the yeast strain *AH109* according to the manufacturer's manual (Clontech).

To confirm the interaction between PIF7 and 14-3-3 proteins, the CDS of *PIF7*, *PIF7(2A)* and *PIF7 (2D)* was cloned into pGADT7 and 14-3-3s were cloned into pGBKT7. Interactions were tested on SD medium without Leu, Trp and His but with 5 mM 3AT.

## Semi-in vivo pull-down assay

Plant materials were ground with liquid nitrogen and re-suspended in extraction buffer (100 mM Tris-HCl [pH 7.5], 300 mM NaCl, 2 mM EDTA, 1% Trion X-100, 10% glycerol, and protease inhibitor cocktail). Protein extracts were centrifuged at 20,000 x g for 10 min, and the resulting supernatant was incubated with pretreated GST-14-3-3 beads for 2 hr. GST was used as a negative control. Beads were re-suspended with SDS-PAGE loading buffer and analyzed by SDS-PAGE and immunoblotting.

## Generation of transgenic plants

For the overexpression of PIF7 fused with GFP, the full-length CDS of *PIF7* and mutated *PIF7* (△, 5A) were cloned into pMDC43 and transformed into a Col-0 background. To generate transgenic plants that overexpressed mutated PIF7 in the *pif7-1* background, the mutated PIF7(2A) and PIF7 (2D) were created in a plasmid of 35S::*PIF7-Flash* using site-directed mutagenesis. All the constructs were transformed into *Agrobacterium* GV3101. The primers are listed in *Supplementary file 1*.

## Hypocotyl measurements

Quantitative measurements of hypocotyls were performed on scanned images of seedlings using ImageJ software. For measurements of mutants and stable transgenic lines, at least 20 seedlings were used per treatment or genotype. For hypocotyl analysis of T1 transgenic lines under shade, the numbers of seedlings with elongated hypocotyls were counted.

## Transient transformation

*Agrobacterium* cells (GV3101) containing 35S::*GFP-PIF7*, 35S::*GFP-PIF7(5A)*, 35S::*GFP-PIF7(2A)*, 35S::*GFP-PIF7(2D)*, 35S::*GFP-PIF7(5D)*, 35S::*GFP-PIF7△* or BiFC expression vectors were re-

suspended in the induction medium (10 mM MES buffer [pH 5.6], 10 mM MgCl$_2$ and 200 μM aceto-syringone) and were infiltrated into young leaves of 4-week-old tobacco plants. The expression of various fluorescent proteins was analyzed using confocal microscopy or western blotting 36 hr after infiltration.

## Gene expression analysis by quantitative real-time RT-PCR

Total RNA was extracted using an RNApre Plant Kit (TIANGEN, China), and the first-strand cDNA was synthesized using a FastQuant RT kit (with gDNase) (TIANGEN, China). Real-time PCR was performed with a Biorad CFX Connect system. All of the oligonucleotide primers were listed in *Supplementary file 1*.

## Antibodies

Anti-Myc (Cat# M4439 RRID: AB_439694) and anti-FLAG (Cat# F3165 RRID: AB_259529) were purchased from Sigma-Aldrich (USA). Anti-H3 (Cat# AS10 710 RRID: AB_10750790), anti-RbcL (Cat# AS03 037–200 RRID: AB_2175288) and anti-14-3-3 (Cat# AS12 2119 RRID: AB_2619715) were from Agrisera (Sweden). Anti-GFP (Cat# MMS-118P-200 RRID: AB_10063778) was from Covance (USA).

# Acknowledgements

We thank Dr. Hongquan Yang (Fudan University) for sharing the cDNA library for yeast two hybrid screens, Dr. Yan Guo (China Agricultural University) and Dr. Honghong Hu (Huazhong Agricultural University) for sharing seeds of 14-3-3 mutants and 14-3-3 λ transgenic lines, Dr. Xuelu Wang (Huazhong Agricultural University) for sharing seeds of BES1-GFP and Dr. Liang Cai (Fudan University) for technical assistance relating to spanning-disk confocal microscopy. Dr. Yanhong Li and Miss Lin Huang at Proteomics Platform (Fudan University) are acknowledged for their help in MS characterization of phosphor-sites of PIF7. This work was supported by National Key R and D Program of China (grant 2017YFA0503800) and by the National Natural Science Foundation of China (grants 31470374 and 31500973).

# Additional information

### Funding

| Funder | Grant reference number | Author |
|---|---|---|
| National Natural Science Foundation of China | 31470374 | Lin Li |
| National Natural Science Foundation of China | 31500973 | Lin Li |
| National Key Research and Development Program of China | 2017YFA0503800 | Lin Li |

The funders had no role in study design, data collection and interpretation, or the decision to submit the work for publication.

### Author contributions

Xu Huang, Conceptualization, Data curation, Formal analysis, Validation, Investigation, Visualization, Methodology; Qian Zhang, Conceptualization, Formal analysis, Validation, Investigation, Methodology; Yupei Jiang, Chuanwei Yang, Formal analysis, Validation, Investigation, Methodology; Qianyue Wang, Validation; Lin Li, Conceptualization, Resources, Data curation, Formal analysis, Supervision, Funding acquisition, Investigation, Methodology, Writing—original draft, Project administration, Writing—review and editing

### Author ORCIDs

Xu Huang http://orcid.org/0000-0002-2711-9920
Lin Li http://orcid.org/0000-0003-4840-5245

Decision letter and Author response
Decision letter https://doi.org/10.7554/eLife.31636.027
Author response https://doi.org/10.7554/eLife.31636.028

## Additional files

### Supplementary files

• Supplementary file 1. Primers used in this study.
DOI: https://doi.org/10.7554/eLife.31636.021

• Transparent reporting form
DOI: https://doi.org/10.7554/eLife.31636.022

### Data availability

All data generated or analysed during this study are included in the manuscript and supporting files.

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
