## [Decision Letter]

Thank you for submitting your article "Shade-induced nuclear localization of PIF7 is regulated by phosphorylation and 14-3-3 proteins in *Arabidopsis*" for consideration by *eLife*. Your article has been favorably evaluated by Christian Hardtke (Senior Editor) and three reviewers, one of whom, Zhi-Yong Wang (Reviewer #1), is a member of our Board of Reviewing Editors.

The reviewers have discussed the reviews with one another and the Reviewing Editor has drafted this decision to help you prepare a revised submission.

Summary:

Overall, the reviewers found your manuscript interesting and potentially a good contribution to *eLife*. However, they also raised important issues that must be addressed. Some of the issues require additional experimental evidence. Particularly, the role of 14-3-3s in controlling PIF7 nuclear localization needs more rigorous tests.

Essential revisions:

1) PIF7 is known to mediate shade avoidance syndrome (SAS) in *Arabidopsis*. Unlike other PIFs, PIF7 is not degraded rapidly in response to phytochrome-induce phosphorylation. How phosphorylation affects PIF7 activity has remained an outstanding question. Therefore, your findings that 14-3-3 proteins retain phospho-PIF7 in cytoplasm and shade-induced dephosphorylation causes PIF7 nuclear localization answer an important question in plant biology. While the results are interpreted clearly, the biological implication was not fully elaborated. For example, how such mechanism may contribute to the rapid response to shade, compared to the other light-labile PIFs that would require new protein synthesis to regain activity upon shade inactivation of phytochrome? We suggest you add a discussion of the possible implication of this nuclear localization mechanism in the contexts of the degradation mechanisms of other PIFs as well as your earlier finding that *pif7* lacks an early shade avoidance response (Li et al., 2012).

2) Two previous studies on GFP/CFP-tagged PIF7 (Leivar et al., 2008 and Kidokoro et al., Plant Physiol., 2009) did not report such cytoplasmic localization. Instead, GFP-PIF7 was shown to be nuclear and in speckles in light-treated tissues. The possible reasons for the discrepancy must be discussed, and ideally the possible reasons should be tested experimentally.

3) Why are BiFC signals stronger in the nucleus than in cytoplasm? If the interaction causes cytoplasmic retention, BiFC signals should be weaker or absent in the nucleus. Co-expression of a PIF7-RFP would help determine whether the tethering to 14-3-3 by BiFC affects PIF7 localization. Comparing the localization patterns of PIF7/14-3-3 BiFC signal with PIF7-GFP after short time shade treatment will provide further evidence as to whether change of 14-3-3 binding is required for shade-induced nuclear localization.

4) The R18 peptide treatment and 14-3-3 mutant experiments did not support your model. Unlike mutations of 14-3-3 binding site in PIF7 (PIF7(2A) and PIF7(5A)), both R18 treatment and 14-3-3 mutations had no effects on the PIF7 localization or target gene expression under white light. They only slightly increased the N/C ration after shade treatment. This needs to be resolved. Perhaps a positive control (another protein whose subcellular localization affected by 14-3-3 binding) is required to make sure the R18 treatment is effective, and higher order 14-3-3 mutants might show more dramatic effects. Similarly, the effects of R18 and 14-3-3 mutations on PIF7 phosphorylation status are weak and variable (Figure 4C and 4F), whereas PIF7(2A) had a strong effect on dephosphorylation under white light conditions (Figure 3C). The quantitative change of phosphorylation of PIF7(2A) (Figure 3C) suggests an indirect effect of the mutations on phosphorylation of additional sites, which may contribute to the change of nuclear localization and phenotypes.

5) The 14-3-3 binding to PIF7(2D) and PIF7(5D) needs to be tested quantitatively in vitro and in vivo, and the phenotypes of plants expressing PIF7(2D) and PIF7(5D) should be analyzed. It has been reported that 14-3-3 binding requires phosphorylation and cannot be mimicked by S-to-D substitution. If PIF7(5D) cannot bind to 14-3-3s, it would suggest that cytoplasmic localization is independent of 14-3-3, though dependent on phosphorylation (negative charge). In this case, it would be important to test whether PIF7(2D) and PIF7(5D) rescue the *pif7* mutant and change localization in response to shade, and you may want to test whether 14-3-3 inhibits PIF7 through other mechanisms such as DNA-binding or transcription activity.

6) Overexpression from the 35S promoter was used for both subcellular localization and in vivo interaction experiments. Reviewers are concerned that overexpression may contribute to artifacts on both cytoplasmic localization and interaction with 14-3-3 proteins. It is important to repeat these experiments using PIF7 expressed from its native promoter at its normal level.

7) The interaction between PIF7 and 14-3-3 should be analyzed by co-immunoprecipitation, or quantitative IP-MS (Figure 2—figure supplement 1 was not quantitative), in *Arabidopsis* under light and shade conditions.

[Editors' note: further revisions were requested prior to acceptance, as described below.]

Thank you for resubmitting your work entitled "Shade-induced nuclear localization of PIF7 is regulated by phosphorylation and 14-3-3 proteins in *Arabidopsis*" for further consideration at *eLife*. Your revised article has been favorably evaluated by Christian Hardke (Senior Editor) and three reviewers, one of whom is a member of our Board of Reviewing Editors.

The manuscript has been improved but there are some remaining issues that need to be addressed before acceptance, as outlined below. Ordinarily we do not encourage multiple rounds of review. Given the remaining substantial concerns, it will be necessary to limit this to just a single additional review before we make a binding final decision on your manuscript.

The main issues are raised by your new data showing that the PIF7(2D) mutation abolished the binding to 14-3-3 λ and 14-3-3 κ, but unexpectedly lead to cytoplasmic retention of PIF7.

1) The Discussion of the role of 14-3-3s must be consistent with the results. There are overstatements about this in the current manuscript.

2) Stronger evidence is needed to clarify the role of 14-3-3s in regulating PIF7. As the effects of R18 treatment and 14-3-3 mutations are subtle, the results need to be quantified with at least three biological replicates. Further, treatment of the 14-3-3 mutants with R18 and R18(Lys) should be tested for possibly stronger effects on the dephosphorylation and nuclear localization of PIF7.

3) Is it possible that PIF7(2D) binds to some other 14-3-3 family members in vivo? One test of this possibility is to immunoprecipitate PIF7 and PIF7(2D) followed by immunoblotting using the anti-14-3-3 antibodies, which may detect other isoforms. Another test is IP-MS as explained below.

4) The IP-MS experiment (Figure 2—figure supplement Figure 1) is invalid due to the lack of a negative control. The IP-MS experiment needs to be re-performed with a negative control using quantitative mass spectrometry to show specificity. If PIF7(2D) is included and compared to wild type PIF7 under the shade condition or compared to PIF7(2A), the protein that mediates cytoplasmic retention of PIF7(2D) can potentially be identified.

*Reviewer #1:*

The revision has improved the manuscript. Particularly, the co-IP experiment (Figure 2DFigure) provides convincing evidence for in vivo interaction that is light/shade-dependent.

However, the new data showing the lack of 14-3-3 binding to PIF7(2D) suggests that binding by 14-3-3 is not essential for cytoplasmic retention and inactivation of PIF7, and additional 14-3-3-independent mechanisms may exist. This should be clearly discussed. The statement "Here we demonstrate… an essential role of the 14-3-3 proteins in cytoplasmic-retention of PIF7 in Arabidopsis" is no longer supported by the new data.

The mass spectrometry data (Figure 2—figure supplement 1) lacks a proper negative control and thus is invalid. Such IP-MS often detects many non-specific background proteins, and a negative control is essential to get meaningful results. I suggest this data be removed. As the interaction has been confirmed by Y2H, BiFC, co-IP, and site-mutagenesis, removing the MS data does not affect the conclusion while avoiding below-standard results.

*Reviewer #2:*

The authors have addressed most of my comments experimentally and provided explanation for discrepancies.

The only concern that I have is the lack of proof that the epitope-tagged lines of PIF7 used in the study is functional (i.e. if they can rescue *pif7* mutants under native conditions). Though, as the authors have argued with other supporting data and time constrain, this may be addressed in future studies.

*Reviewer #3:*

The revised manuscript by Huang et al. addressed some issues raised by reviewers, but the functional roles of 14-3-3 proteins in the regulation of PIF7 phosphorylation and localization are still not clear and not supported by experimental evidence. One of major molecular functions of 14-3-3 proteins is to retain their target proteins in the cytoplasm, thereby preventing them from translocating to the nucleus. However, several experimental evidences (see below) suggest that the 14-3-3 proteins are not directly involved in the PIF7 localization although they seem to directly bind to the phosphorylated PIF7.

1) The 14-3-3 proteins interact with PIF7 specifically under white light (not shade).

2) Neither R18 treatment nor 14-3-3 mutations affected the PIF7 localization under white light.

3) PIF7(2D) does not interact with the 14-3-3 proteins, but it is mainly localized in the cytoplasm.

Since the cytoplasmic localization of PIF7 is not dependent on the 14-3-3 proteins (as the authors also suggested in "Responses to reviewers' comments"), the manuscript (especially Discussion part) should be re-written to clearly define the role of 14-3-3 proteins. The title should be changed; the 14-3-3 proteins only slightly affect the PIF7 localization in the very limited condition.

In addition, although authors claim the 14-3-3 proteins somehow affect the shade-induced PIF7 localization and dephosphorylation, the effects of R18 and 14-3-3 mutations on the PIF7 localization are very subtle (Figure 4A, D) and their effects on the dephosphorylation of PIF7 are not obvious (Figure 4C, F). Since these are the only effects of 14-3-3 proteins on PIF7 that the authors showed in this manuscript, they should provide more convincing data.

The 14-3-3 proteins interact with PIF7 only under white light, but they appear to regulate the PIF7 localization only during the transition from white to shade. Possible explanations for this inconsistency should be provided in Discussion part.

---

## [Author Response]

Essential revisions:1) PIF7 is known to mediate shade avoidance syndrome (SAS) in Arabidopsis. Unlike other PIFs, PIF7 is not degraded rapidly in response to phytochrome-induce phosphorylation. How phosphorylation affects PIF7 activity has remained an outstanding question. Therefore, your findings that 14-3-3 proteins retain phospho-PIF7 in cytoplasm and shade-induced dephosphorylation causes PIF7 nuclear localization answer an important question in plant biology. While the results are interpreted clearly, the biological implication was not fully elaborated. For example, how such mechanism may contribute to the rapid response to shade, compared to the other light-labile PIFs that would require new protein synthesis to regain activity upon shade inactivation of phytochrome? We suggest you add a discussion of the possible implication of this nuclear localization mechanism in the contexts of the degradation mechanisms of other PIFs as well as your earlier finding that pif7 lacks an early shade avoidance response (Li et al., 2012).

We have followed the reviewer’s suggestion by including the related discussion in our revised manuscript (Discussion, second and fifth paragraphs).

2) Two previous studies on GFP/CFP-tagged PIF7 (Leivar et al., 2008 and Kidokoro et al., Plant Physiol., 2009) did not report such cytoplasmic localization. Instead, GFP-PIF7 was shown to be nuclear and in speckles in light-treated tissues. The possible reasons for the discrepancy must be discussed, and ideally the possible reasons should be tested experimentally.

As the reviewer mentioned, GFP/CFP-tagged PIF7 has been published in two previous studies.

According to the phenotype of loss function of mutants (*pif7-1, pif7-2*) (Leivar et al., 2008, Li et al., 2012), PIF7 is a positive regulator for hypocotyl elongation. In our hand, overexpressing either PIF7-Flash (Li., et al., 2012 and current study) or N-terminated tagged GFP-PIF7 (current study) could rescue the shorter hypocotyl phenotype of *pif7-1* or *pif7-2* under shade. So we believe that the PIF7-Flash or GFP-PIF7 transgenic lines represent its normal function and probable localization. The line reported in literature, PIF7-CFP overexpression lines from Dr. Peter Quail’s lab show a somewhat short hypocotyl phenotype in white light and dark (Leivar et al., 2008), indicating C-terminate fused GFP protein might interfere the normal function of PIF7. It is a possible explanation of the discrepancy of these localization results.

3) Why are BiFC signals stronger in the nucleus than in cytoplasm? If the interaction causes cytoplasmic retention, BiFC signals should be weaker or absent in the nucleus. Co-expression of a PIF7-RFP would help determine whether the tethering to 14-3-3 by BiFC affects PIF7 localization. Comparing the localization patterns of PIF7/14-3-3 BiFC signal with PIF7-GFP after short time shade treatment will provide further evidence as to whether change of 14-3-3 binding is required for shade-induced nuclear localization.

The seemly stronger nuclear signals in BiFC assay is probably because these intensive signals gathered in a small area in tobacco leaf cells. In *Arabidopsis*, GFP-PIF7 localized in both nucleus and cytoplasm (Figure 1), but 14-3-3 λ mainly localized in cytoplasm (Figure 6—figure supplement 1). To further clarify this question, we quantitated the nuclear/cytoplasmic fluorescence signal ratio in tobacco leaf cells which co-expressed 14-3-3λ-cYFP and nYFP-PIF7 or only expressed GFP-PIF7. As shown in Author response image 1, nuclear/cytoplasmic signal is significant lower in BiFC assay, indicating that the interaction of 14-3-3s and PIF7 occurs mainly in cytoplasm.

The experiment suggested by the reviewers is to figure out whether change of 14-3-3 binding is required for shade-induced nuclear localization of PIF7. However, the localization of GFP-PIF7 expressed in tobacco leaf cells doesn’t respond to shade treatment (Author response image 1). The shade response in tobacco leaf cells may not share the same molecular mechanism as that in *Arabidopsis.* Regardless of light response, we could test whether co-expression of 14-3-3s affects the localization of PIF7 in tobacco leaf cells. When 14-3-3 λ was co-expressed with GFP-PIF7, GFP-PIF7(2A), GFP-PIF7(5A), GFP-PIF7(2D), GFP-PIF7(5D), or GFP in tobacco leaves (Author response image 1), 14-3-3λ was unable to change the localization of any version of PIF7. Moreover, *14-3-3 λκ* mutation and treatment of R18 didn’t change the localization of GFP-PIF7 under white light in *Arabidopsis* (Figure 4). Taken together, tobacco system was used to determine the interaction of 14-3-3s and PIF7 preferentially in cytoplasm and localization of PIF7 derivatives, whereas *Arabidopsis* system is used to address the questions of shade and phosphorylation dependency.

**Author response image 1. respfig1:** (**a**) Quantification of ratios between nuclear and cytoplasmic intensities of YFP/GFP in tobacco leaf cells (n>10) which co-expressed 14-3-3λ-cYFP and nYFP-PIF7 or only expressed GFP-PIF7. (**b**) Shade light doesn’t affect the localization of GFP-PIF7 expressed in tobacco leaf cells. (**c**) Co-expressed 14-3-3λ doesn’t affect the localization of PIF7 derivatives in tobacco leaf cells. Subcellular localization of GFP, GFP-PIF7, GFP-PIF7 (2A), GFP-PIF7 (5A), GFP-PIF7 (2D) and GFP-PIF7 (5D) co-expressed with 14-3-3λ or empty vector in tobacco leaf cells. The expression levels of GFP-PIF7 derivatives and 14-3-3λ-FLAG was detected using anti-GFP antibody and anti-FLAG antibody, respectively.

4) The R18 peptide treatment and 14-3-3 mutant experiments did not support your model. Unlike mutations of 14-3-3 binding site in PIF7 (PIF7(2A) and PIF7(5A)), both R18 treatment and 14-3-3 mutations had no effects on the PIF7 localization or target gene expression under white light. They only slightly increased the N/C ration after shade treatment. This needs to be resolved. Perhaps a positive control (another protein whose subcellular localization affected by 14-3-3 binding) is required to make sure the R18 treatment is effective, and higher order 14-3-3 mutants might show more dramatic effects. Similarly, the effects of R18 and 14-3-3 mutations on PIF7 phosphorylation status are weak and variable (Figure 4C and 4F), whereas PIF7(2A) had a strong effect on dephosphorylation under white light conditions (Figure 3C). The quantitative change of phosphorylation of PIF7(2A) (Figure 3C) suggests an indirect effect of the mutations on phosphorylation of additional sites, which may contribute to the change of nuclear localization and phenotypes.

We thank the reviewers for pointing out these issues. Actually, our model proposed that 14-3-3s retain the phosphorylated PIF7 in cytoplasm under shade, possibly through protection from phosphatases. Our results (Figure 5) support 14-3-3s mainly work during the transition from phosphorylated to dephosphorylated PIF7, not under white light. If 14-3-3s’ function is to escort phosphorylated PIF7 to cytoplasm, we should have observed longer hypocotyl in white light in which phosphorylated PIF7 is dominant, and more cytoplasmic localization of PIF7 in 14-3-3s mutant background than that in Col-0 under white light. But in fact, neither of them happens.

The effect of R18 is significant on shade-induced nuclear-translocation and dephosphorylation of PIF7 (Figure 5). Moreover, the effect of R18 treatment on shade-induced gene expression and hypocotyl elongation reproduces that of 14-3-3 mutations (Figure 6). So we believe that R18 treatment is effective. We also followed the reviewer’s suggestion and tested the effect of R18 on localization of BES1-GFP, as shown in Author response image 2. The subcellular localization of BES1 is affected by 14-3-3 binding (Dev Cell. 2007 13(2):177-89; Mol. Cells. 2010 29:283-290) and also affected by R18 treatment.

Six 14-3-3 proteins interacted with PIF7. There are thirteen 14-3-3 proteins in *Arabidopsis*, which lead to functional abundance and difficulty to study loss of function of 14-3-3s. Although we agree with the reviewer that higher order 14-3-3-mutants might show more dramatic effects, it’s technically very challenging to construct a higher order mutant with a lot of isoforms. On the other hand, the double mutant of 14-3-3λκ significantly increased the N/C ratio after shade treatment, supporting our hypothesis of 14-3-3s’ effect on PIF7.

We agree with the reviewers that there are additional sites beside S139 and S141 that may contribute to the change of nuclear localization and phenotypes.

**Author response image 2. respfig2:** The effect of R18 on the localization of BES1-GFP. BES1-GFP transgenic plants (a gift from Dr. Xuelu Wang’s lab) grown under white light were treated with R18 or R18(Lys) for 3 hr after 10 min of vacuum. ImageJ was used to quantify the fluorescence intensities. Ratios between the nuclear and cytoplasmic signal intensities were calculated from at least 10 cells for each treatment.

5) The 14-3-3 binding to PIF7(2D) and PIF7(5D) needs to be tested quantitatively in vitro and in vivo, and the phenotypes of plants expressing PIF7(2D) and PIF7(5D) should be analyzed. It has been reported that 14-3-3 binding requires phosphorylation and cannot be mimicked by S-to-D substitution. If PIF7(5D) cannot bind to 14-3-3s, it would suggest that cytoplasmic localization is independent of 14-3-3, though dependent on phosphorylation (negative charge). In this case, it would be important to test whether PIF7(2D) and PIF7(5D) rescue the pif7 mutant and change localization in response to shade, and you may want to test whether 14-3-3 inhibits PIF7 through other mechanisms such as DNA-binding or transcription activity.

We have now included the data about *PIF7(2D)-Flash* transgenic line in *pif7-1* background. As shown in Figure 4C, E, F and G of our revised manuscript, PIF7(2D)-Flash mainly localizes in cytoplasm and can’t complement the shade defective gene expression and phenotype of *pif7-1*, indicating that S-to-D substitution could mimic phosphorylation of PIF7. However, in our BiFC and GST pull-down assay, PIF7(2D) can’t interact with 14-3-3s (Figure 3—figure supplement 3), probably because the interaction of 14-3-3s and PIF7 can’t be mimicked by S-to-D substitution as the reviewer mentioned. So we would like to conclude that the cytoplasmic localization of PIF7 is dependent on the phosphorylation state, not 14-3-3s under white light. But 14-3-3s delay the shade-induced nuclear- translocation and dephosphorylation of PIF7 (Figure 5), possibly through protection from phosphatases during the transition from white light to shade. We did test the effect of 14-3-3s on PIF7’s transcriptional activity in tobacco leaves as the reviewer suggested. As we expected, co-expressed 14-3-3λ or 14-3-3κ didn’t change the activation of PIF7 on *YUCCA8* promoter, indicating the inhibitions of 14-3-3λ and 14-3-3κ on PIF7 are not through the regulation of the transcriptional activity (Author response image 3).

**Author response image 3. respfig3:** The effect of 14-3-3λ and 14-3-3κ on PIF7-mediated activation of *YUCCA8* promoter. The effector constructs contain the CaMV 35S promoter fused to the transcription factor PIF7 or 14-3-3s. The reporter construct contains the 2.2 Kb upstream of the translation initiation site of *YUCCA8* fused to the LUC reporter gene. Both effector and reporter were co-expressed in tobacco leaf cells. The ratio of Luc to Ren from leaves transfected reporter and empty effector was normalized as 1.

6) Overexpression from the 35S promoter was used for both subcellular localization and in vivo interaction experiments. Reviewers are concerned that overexpression may contribute to artifacts on both cytoplasmic localization and interaction with 14-3-3 proteins. It is important to repeat these experiments using PIF7 expressed from its native promoter at its normal level.

We have been working on generating native promoter driven PIF7 expressed transgenic lines for a while. We amplified 1019 bp before ATG of PIF7 as native promoter according to the published paper (Kidokoro et al., Plant Physiol., 2009) and constructed pPIF7::*PIF7-Flash*. But unfortunately, we didn’t get complementary line during the screen of pPIF7::*PIF7-Flash* transgenic lines in *pif7-1* background so far. The length of PIF7 promoter we used might be not long enough. Although we agree with the reviewer better to repeat the localization experiment using PIF7 expressed from its native promoter, we need more studies on the length of native promoter before we use it for further study.

However, overexpressed PIF7(2A) and PIF7(2D) displayed the different localization compared to the overexpressed PIF7, indicating the effect of localization is not an artifact of overexpression. Interactions between PIF7 and 14-3-3s have been verified in Y2H, BiFC, GST-pulldown assays and CoIP. We also noticed that overexpression lines have been used for several 14-3-3s’ clients related studies [Dev Cell. 2007 13(2):177; Dev Cell. 2011 21(5):825; Plant Cell. 2014, 26(3):1166]. So we feel confident about our conclusion based on meaningful data.

7) The interaction between PIF7 and 14-3-3 should be analyzed by co-immunoprecipitation, or quantitative IP-MS (Figure 2—figure supplement 1 was not quantitative), in Arabidopsis under light and shade conditions.

We have followed the reviewer’s suggestion and did Co-IP experiment. The data confirmed the binding of 14-3-3s to the phosphorylated PIF7 (revised Figure 2D).

[Editors' note: further revisions were requested prior to acceptance, as described below.]

The main issues are raised by your new data showing that the PIF7(2D) mutation abolished the binding to 14-3-3 λ and 14-3-3 κ, but unexpectedly lead to cytoplasmic retention of PIF7.

Thanks for raising this issue. We hope our newly included data will convince the reviewers of our conclusions.

1) The Discussion of the role of 14-3-3s must be consistent with the results. There are overstatements about this in the current manuscript.

We have modified the Discussion (third and fourth paragraphs).

2) Stronger evidence is needed to clarify the role of 14-3-3s in regulating PIF7. As the effects of R18 treatment and 14-3-3 mutations are subtle, the results need to be quantified with at least three biological replicates. Further, treatment of the 14-3-3 mutants with R18 and R18(Lys) should be tested for possibly stronger effects on the dephosphorylation and nuclear localization of PIF7.

Effects of R18 treatment and 14-3-3 mutations are significant on regulating PIF7’s localization and de-phosphorylation. Quantification of the shade-induced nuclear accumulation of GFP-PIF7 was calculated from at least 10 cells that came from different seedlings (Figure 5B and E). Biological replicates have been newly added in Figure 4C and F to further confirm the effects on de-phosphorylation of PIF7. We also quantitated the ratio of phosphorylated PIF7 to total PIF7 and presented in Figure 5—figure supplement 2. The shade-induced de-phosphorylation of PIF7 is significantly differentiated between R18 and R18 (Lys) treatment, and the effect is even stronger between 14-3-3 mutant and Col-0.

Thanks for the suggestion to treat 14-3-3 mutants with R18. Shade-induced nuclear translocation and de-phosphorylation of PIF7 is a fast process, which is correlated with the quick changes of gene expression and the measurable growth after just 30-40 min shade exposure. The effects of R18 and 14-3-3 mutations accelerate this fast process. We have shown the significant effects by individual treatment of R18 or in 14-3-3 mutants. But to capture the probable stronger effects by both factors have been very challenging due to the handling time and resolution of detective methods.

3) Is it possible that PIF7(2D) binds to some other 14-3-3 family members in vivo? One test of this possibility is to immunoprecipitate PIF7 and PIF7(2D) followed by immunoblotting using the anti-14-3-3 antibodies, which may detect other isoforms. Another test is IP-MS as explained below.

Thanks for this suggestion. We tried Co-IP in vivo and Y2H experiments. The new data in Figure 3A and C shows that 14-3-3s bind to the PIF7 (2D), not PIF7 (2A) in in vivo and in the yeast system. The antibody we used is not isoform specific, so we cannot distinguish 14-3-3 family members yet.

4) The IP-MS experiment (Figure 2—figure supplement 1) is invalid due to the lack of a negative control. The IP-MS experiment needs to be re-performed with a negative control using quantitative mass spectrometry to show specificity. If PIF7(2D) is included and compared to wild type PIF7 under the shade condition or compared to PIF7(2A), the protein that mediates cytoplasmic retention of PIF7(2D) can potentially be identified.

We totally agree with the reviewer. We did IP-MS, using the different light treated transgenic seedlings, when we started the project, to qualitatively identify PIF7’s binder. But soon a substantial amount of evidence revealed the interaction of PIF7 and 14-3-3s. Therefore, for the sake of clarity and concision, we would rather remove the 14-3-3’s MS data and keep the identification of phosphorylated sites from IP-MS in our revised manuscript.

It is a fascinating idea to compare the partners in PIF7(2A) or PIF7(2D) complex using IP-MS. Our new Co-IP experiment has shown the different levels of 14-3-3s bound with PIF7, PIF(2A) and PIF7(2D). We believe that there are more components besides 14-3-3s involved in the translocation of PIF7, such as kinase and phosphatase. The on-going ID of these components would likely greatly forward our understanding of PIF7-related shade-signaling pathway, but to characterize these components requires substantial work and deserves a separate study.

Reviewer #1:The revision has improved the manuscript. Particularly, the co-IP experiment (Figure 2D) provides convincing evidence for in vivo interaction that is light/shade-dependent.However, the new data showing the lack of 14-3-3 binding to PIF7(2D) suggests that binding by 14-3-3 is not essential for cytoplasmic retention and inactivation of PIF7, and additional 14-3-3-independent mechanisms may exist. This should be clearly discussed. The statement "Here we demonstrate… an essential role of the 14-3-3 proteins in cytoplasmic-retention of PIF7 in Arabidopsis" is no longer supported by the new data.

The new data in Figure 3A and C shows that 14-3-3s bind to the PIF7 (2D), not PIF7 (2A), in in vivo and in the yeast system. The related sentence has been modified in the last paragraph of the Introduction.

The mass spectrometry data (Figure 2—figure supplement 1) lacks a proper negative control and thus is invalid. Such IP-MS often detects many non-specific background proteins, and a negative control is essential to get meaningful results. I suggest this data be removed. As the interaction has been confirmed by Y2H, BiFC, co-IP, and site-mutagenesis, removing the MS data does not affect the conclusion while avoiding below-standard results.

We totally agree with the reviewer. We did IP-MS, using the different light treated transgenic seedlings, when we started the project, to qualitatively identify PIF7’s binder. But soon a substantial amount of evidence revealed the interaction of PIF7 and 14-3-3s. Therefore, for the sake of clarity and concision, we would rather remove the 14-3-3’s MS data and keep the identification of phosphorylated sites from IP-MS in our revised manuscript.

Reviewer #3:The revised manuscript by Huang et al. addressed some issues raised by reviewers, but the functional roles of 14-3-3 proteins in the regulation of PIF7 phosphorylation and localization are still not clear and not supported by experimental evidence. One of major molecular functions of 14-3-3 proteins is to retain their target proteins in the cytoplasm, thereby preventing them from translocating to the nucleus. However, several experimental evidences (see below) suggest that the 14-3-3 proteins are not directly involved in the PIF7 localization although they seem to directly bind to the phosphorylated PIF7.1) The 14-3-3 proteins interact with PIF7 specifically under white light (not shade).2) Neither R18 treatment nor 14-3-3 mutations affected the PIF7 localization under white light.

We understand the paradoxical discrepancy of interaction and function in white light and have been working on this question from the beginning.

Under our white light conditions, there is no significant effect of R18 and 14-3-3 mutations on PIF7 localization and de-phosphorylation in Figure 5. Together with the hypocotyl length in our white light (Figure 6), we do not claim the role of 14-3-3s under white light. We think the cytoplasmic localization of PIF7 is dependent on the phosphorylation, possibly not dependent on 14-3-3s. However, these results could be influenced by 1) strong white light radiance; 2) the low potency of R18 as an inhibitor of 14-3-3s and 3) the redundancy of thirteen 14-3-3s in *Arabidopsis*. To clearly demonstrate the role of 14-3-3s under white light, to our knowledge so far, it is necessary to obtain, at least, high order 14-3-3 mutants, stronger inhibitor without side-effect and change the light condition. In contrast, 14-3-3s significantly delay the shade-induced translocation and de-phosphorylation of PIF7 (Figure 5), and consequently enhance shade-induced hypocotyl elongation. The weak shade phenotype might be caused by functional redundancy of 14-3-3 proteins. It’s also possible that compensatory increase of other isoforms occurs in 14-3-3 λκ. 14-3-3 proteins sequester phosphorylated PIF7 in cytoplasm possibly through protection from phosphatases. All these experimental results support the role of 14-3-3s through the interaction with PIF7 during the transition from white light to shade. We have added the related discussion (Discussion, fourth paragraph) to the revised version.

3) PIF7(2D) does not interact with the 14-3-3 proteins, but it is mainly localized in the cytoplasm.

We conducted Co-IP and Y2H experiments and showed that 14-3-3s bind to the PIF7 (2D). Actually, more 14-3-3 proteins were co-immunoprecipitated with PIF7 (2D) than that with PIF7 (2A) in *Arabidopsis* seedlings, which is consistent with the cytoplasmic retention of PIF7 (2D).

Since the cytoplasmic localization of PIF7 is not dependent on the 14-3-3 proteins (as the authors also suggested in "Responses to reviewers' comments"), the manuscript (especially Discussion part) should be re-written to clearly define the role of 14-3-3 proteins. The title should be changed; the 14-3-3 proteins only slightly affect the PIF7 localization in the very limited condition.

The major and novel findings in our work include at least 4 points: 1) shade rapidly induce nuclear localization of PIF7; 2) Phosphorylation sites of PIF7 are important for its localization and function; 3) 14-3-3s interact with PIF7; 4) 14-3-3s delay the nuclear translocation and de-phosphorylation of PIF7 and negatively regulate SAS. These findings are supported by experimental evidences and appropriately reflected in title: “Shade-induced nuclear localization of PIF7 is regulated by phosphorylation and 14-3-3 proteins in *Arabidopsis*”.

In addition, although authors claim the 14-3-3 proteins somehow affect the shade-induced PIF7 localization and dephosphorylation, the effects of R18 and 14-3-3 mutations on the PIF7 localization are very subtle (Figure 4A, D) and their effects on the dephosphorylation of PIF7 are not obvious (Figure 4C, F). Since these are the only effects of 14-3-3 proteins on PIF7 that the authors showed in this manuscript, they should provide more convincing data.

Shade-induced nuclear translocation and de-phosphorylation of PIF7 is a fast process, which is correlated with the quick changes of gene expression and measurable growth after just 30-40 min shade exposure. The effects of R18 and 14-3-3 mutations accelerate this fast process. Based on Figures 5B, 5E and newly added Figure 5—figure supplement 2, significant effects of R18 and 14-3-3 mutations have been quantified with more experimental repeats.